# The Use of *Nigella sativa* in Cardiometabolic Diseases

**DOI:** 10.3390/biomedicines12020405

**Published:** 2024-02-09

**Authors:** Giuseppe Derosa, Angela D’Angelo, Pamela Maffioli, Laura Cucinella, Rossella Elena Nappi

**Affiliations:** 1Department of Internal Medicine and Therapeutics, University of Pavia, 27100 Pavia, Italy; angeladangelo@unipv.it; 2Centre of Diabetes, Metabolic Diseases, and Dyslipidemias, University of Pavia, 27100 Pavia, Italy; p.maffioli@smatteo.pv.it; 3Regional Centre for Prevention, Surveillance, Diagnosis and Treatment of Dyslipidemias and Atherosclerosis, Fondazione IRCCS Policlinico San Matteo, 27100 Pavia, Italy; 4Italian Nutraceutical Society (SINut), 40100 Bologna, Italy; 5Laboratory of Molecular Medicine, Fondazione IRCCS Policlinico San Matteo, University of Pavia, 27100 Pavia, Italy; 6Department of Clinical, Surgical, Diagnostic and Pediatric Sciences, University of Pavia, 27100 Pavia, Italy; l.cucinella@smatteo.pv.it (L.C.); r.nappi@smatteo.pv.it (R.E.N.); 7Research Center for Reproductive Medicine and Gynecological Endocrinology, Menopause Unit, Fondazione IRCCS Policlinico San Matteo, 27100 Pavia, Italy

**Keywords:** *Nigella sativa*, thymoquinone, pharmacological properties

## Abstract

*Nigella sativa* L. is an herb that is commonly used in cooking and in traditional medicine, particularly in Arab countries, the Indian subcontinent, and some areas of eastern Europe. *Nigella sativa* is also called “black cumin” or “black seeds”, as the seeds are the most-used part of the plant. They contain the main bioactive component thymoquinone (TQ), which is responsible for the pleiotropic pharmacological properties of the seeds, including anti-oxidant, anti-inflammatory, anti-hypertensive, anti-hepatotoxic, hypoglycemic, and lipid-lowering properties. In this narrative review, both the potential mechanisms of action of *Nigella sativa* and the fundamental role played by pharmaceutical technology in optimizing preparations based on this herb in terms of yield, quality, and effectiveness have been outlined. Moreover, an analysis of the market of products containing *Nigella sativa* was carried out based on the current literature with an international perspective, along with a specific focus on Italy.

## 1. Introduction

Cardiovascular diseases are the leading cause of death in the Western world and the leading cause of death, hospitalization, and disability among people with type 2 diabetes mellitus. The incidence of cardiovascular disease in people with diabetes is more than double compared to that observed in people without diabetes. Similarly, the mortality rate after a first myocardial infarction in diabetic patients is much higher than that of non-diabetics. *Nigella sativa* L., commonly known as “black cumin” or “black seeds”, is an annual herbaceous hermaphrodite plant belonging to the Ranunculaceae family [1]. The name Nigella comes from the Latin “niger”, or black, which refers to the color of the seeds, whereas sativa comes from the Latin verb “serere”, to sow, or “sativum”, cultivated [2]. The plant has erect branched stems that empty with age, with a color ranging from green to dark green. It has green leaves and flowers that turn from red to blue with maturity. The fruits of the plant are composed of three to six carpels, and each of them contains seeds whose color turns black after maturity and exposure to air [1].

*Nigella sativa* is mainly distributed in North Africa, the Middle East, Europe, and Asia. It is cultivated in various countries such as Egypt, Iran, Greece, Syria, Albania, Turkey, Saudi Arabia, India, and Pakistan [3]. Although it is not a plant belonging to the Italian culinary tradition, it is also cultivated in the region of Marche, located in Central Italy [4]. Currently, India is the largest producer and exporter of *Nigella sativa*, whereas Brazil, Canada, Colombia, the European Union, Ecuador, Japan, Malaysia, Mexico, South Africa, and USA are the main importers [3].

## 2. Uses of *Nigella sativa*

The most used part of *Nigella sativa* comprises the seeds, which are consumed whole, ground, or are pressed to obtain oil. The seeds have been known for more than 2000 years, both for their therapeutic properties and for their use as a spice, especially in Middle Eastern cuisine [1,2,3].

### 2.1. Medicinal Uses

The seeds of *Nigella sativa* are used as therapeutic agents in different traditional medicinal practices like Arabian medicine and Ayurveda. Its manifold uses have earned *Nigella sativa* seeds the Arabic approbation “Habbat-ul-barakah”, which means “seeds of blessing”. They are used for the treatment of respiratory diseases, such as asthma and bronchitis, gastrointestinal diseases, such as indigestion and diarrhea, as well as in cases of amenorrhea, dysmenorrhea, and skin infections. In Ayurvedic medicine, the oil extracted from *Nigella sativa* seeds is known to be anti-inflammatory, antibacterial, anti-oxidant, and immunomodulatory. It is also used in traditional Chinese medicine as a component of a formulation for the treatment of headaches [5,6].

### 2.2. Culinary and Other Uses

The seeds of *Nigella sativa* have a pungent, bitter taste and aroma and, therefore, are used as a flavoring additive in bread and pickles or added to tea and coffee. The dry-roasted seeds are sometimes utilized to flavor curries, pulses, and vegetables. Ground *Nigella sativa* seeds are usually sprinkled on salads or mixed with honey. The seeds are also used as an insect repellent [5]. Their essential oil extracts are used in beauty products like hair oil, body oil, soaps, and shampoos [3].

### 2.3. Use of Nigella Sativa in Modern Medicine

In the literature, there are many published works on the traditional or historical uses of *Nigella sativa*. The first study on *Nigella sativa* appeared in the scientific literature more than a century ago [7], although a growing interest among the scientific community in its health benefits has occurred only in the last two decades [2]. Research on *Nigella sativa* is ongoing in almost half of the world and has developed most in the Middle East and South Asia, especially in Iran, Egypt, India, and Saudi Arabia [2]. It is believed that the bioactive compounds derived from the seeds, particularly the essential oil, which is rich in thymoquinone (TQ), are responsible for various biological activities, such as anti-oxidant, anti-inflammatory, anti-cancer, hypoglycemic, anti-microbial, anti-nephrotoxic, anti-hepatotoxic, and immunostimulating activities.

It has been observed that the highest percentage of published works are focused on the anti-cancer activity (17%) of *Nigella sativa*. However, a number of equally significant studies have investigated its anti-oxidant (13%), anti-inflammatory (10%), and antibacterial (8%) action, as well as its protective effect in diabetes (10%), kidney diseases (5%), skin diseases (5%), cardiac disorders (5%), dyslipidemia (3%), brain disorders (3%), and hepatic diseases (3%) [2].

## 3. Phytochemical Constituents

The phytochemical composition of *Nigella sativa* is different according to its growing region, maturity stage, processing method, and the extraction technique utilized to obtain the oil. The bioactive phytochemicals of black cumin, mostly derived from the seeds (85%), bark (8%), and sprout (7%), comprise primary and secondary metabolites belonging to different chemical classes including terpenes and terpenoids, phytosterols, alkaloids, tocols, polyphenols, and miscellaneous components. There is also a non-volatile component, represented by flavonoids, phenolic acids, tannins, and fatty acids, and a volatile one, represented by terpenic compounds [2]. In particular, the chemical constituents of *Nigella sativa* L. are terpenes and terpenoids (such as thymoquinone and its derivates); fatty acids (the most abundant fatty acids are linoleic acid, oleic acid, and palmitic acid); phytosterols (β-sitosterol and stigmasterol); alkaloids (such as nigellamines); polyphenols (such as quercitrin and kaempferol); carbohydrates; glycerolipids; phospholipids; vitamins; minerals; and alkane hydrocarbons (Figure 1).

### 3.1. Terpenes and Terpenoids

The terpene and terpenoid family is the major chemical group of black cumin, to which belong TQ and its derivatives, such as carvacrol, 4-terpineol, α-pinene, thymol, t-anethol, thymohydroquinone (THQ), dithymoquinone, p-cymene, sesquiterpene, longifolene, and several other compounds. The wide range of black cumin pharmacological properties is mainly due to quinine components, the most prevalent of which is TQ [1,2].

Thymoquinone is the most abundant bioactive component of the volatile oil of *Nigella sativa* L., whose biological activities are mainly ascribed to this substance [8]. However, the pharmaceutical development of TQ is hindered due to several limitations. Firstly, TQ is characterized by high hydrophobicity and poor solubility in aqueous media, which limit its oral bioavailability. In addition, its high level of lipophilicity causes difficulties in successful formulation and affects the stability of the end product. Moreover, TQ exhibits sensitivity to environmental factors such as light and temperature [9]. On the other hand, it has a low molecular weight, which allows it to cross the blood–brain barrier and thus be used to treat different neurodegenerative and neuropsychiatric diseases, also given its anti-oxidant, immunomodulatory, and anti-viral action [8]. It has been reported that TQ showed rapid elimination and relatively slow absorption after oral administration in a rabbit model [10]. Thymoquinone is 99% bound to plasma proteins, including bovine serum albumin and alpha-1 acid glycoprotein [11,12]. It has been demonstrated that TQ undergoes hepatic biotransformation comprising reduction into dihydrothymoquinone and interacts with cytochrome P450 [12]. The low concentration of TQ in the oil of *Nigella sativa* L. and its poor oral bioavailability resulted in the use of high amounts of oil, between 3 and 10 mL/day, in clinical studies, which could have been responsible for the gastrointestinal side effects. Instead, the use of concentrated extracts has proven to be safe and tolerable. Against this research background, new strategies for the formulation of TQ and newer techniques must be developed, including the use of nanotechnology, in order to improve the bioavailability of TQ without affecting its efficacy and safety [10,13,14].

### 3.2. Phytosterols

Oil extracted from *Nigella sativa* contains several sterols, of which β-sitosterol is the most abundant (44–54%), followed by stigmasterol, whose content is equal to 16.57–20.92% of total sterols [15].

Other sterols, such as ∆7-stigmasterol, ∆7-avenasterol, campesterol, and cholesterol, are present in smaller percentages. *Nigella sativa* can be considered a natural agent capable of lowering blood cholesterol and preventing cardiovascular diseases (CVDs) due to the presence of high levels of sterols [2].

### 3.3. Alkaloids

Alkaloids of *Nigella sativa* can be classified on the basis of the alkaloid skeletons: isoquinoline alkaloids, such as nigellicimine and nigellicimine-N-oxide, and pyrazole or indazole alkaloids, such as nigellidine and nigellicine. Moreover, alkaloid nigelamines A1–A5, belonging to the diterpene family, exhibit strong lipid-metabolism-promoting activity [16].

### 3.4. Fatty Acids

*Nigella sativa* seeds contain different types of fatty acids, among which the most abundant is linoleic acid (55.6%), followed by oleic acid (23.4%) and palmitic acid (12.5%). Other fatty acids such as stearic acid, lauric acid, myristic acid, linolenic acid, and eicosadienoic acid are present in small amounts of between 0.5% and 3.4% [17].

### 3.5. Tocols

Tocopherols are important natural anti-oxidants that scavenge free radicals and inhibit lipid peroxidation in biological membranes. There are four isomers of tocopherols, alpha (α), beta (β), gamma (γ), and delta (δ), which are distinguished by the position of the methyl group on the chromanol ring. Among the various tocopherols, the γ-tocopherol content is the highest, with an amount ranging from 8.57 to 34.23 ppm [18]. Differences in the content of tocopherol isomers may arise due to the methods of extraction and variation in the cultivated areas, maturity periods, and storage conditions [19].

### 3.6. Polyphenols

A total of 19 polyphenols were identified from *Nigella sativa* seeds [20], comprising caftaric acid, gentisic acid, caffeic acid, chlorogenic acid, p-coumaric acid, ferulic acid, sinapic acid, cichoric acid, hyperoside, isoquercitrin, rutin, myricetin, fisetin, quercitrin, quercetin, patuletin, luteolin, kaempferol, and apigenin. Of these, quercetin and kaempferol were the most abundant. As potent anti-oxidant polyphenols, kaempferol prevents oxidative damage of cells and quercetin protects against different diseases such as osteoporosis, lung cancer, and cardiovascular events. Kaempferol also prevents arteriosclerosis by inhibiting LDL oxidation and platelet aggregation [21,22].

### 3.7. Miscellaneous Components

*Nigella sativa* contains several other chemical constituents including protein, carbohydrates (rhamnose, xylose, and arabinose), glycerolipids (monoacylglycerols, diacylglycerols, and triacylglycerols), and phospholipids (phosphatidylinositol, phosphatidylcholine, and phosphatidylglycerol). It also contains triterpene saponin (α-hederin), phenylpropanoids (eugenol, ε-cinnamaldehyde), vitamins (vitamins A, E, and C, folic acid, thiamin, riboflavin, pyridoxine, and niacin), minerals (of which iron is the most abundant), and some alkane hydrocarbons (n-nonane, 2-undecanone, n-octyl isobutyrate, and 8-heptadecene) [1,2,5].

## 4. Pharmacological Activities of *Nigella sativa*

Several pharmaceutical and biological properties have been attributed to *Nigella sativa* and its main active constituent, TQ. Numerous studies have also demonstrated that this plant is medicinally very effective against various diseases, including CVD, diabetes, and metabolic diseases, inflammatory conditions, and menopause.

Several possible mechanisms are involved in the protective cardiometabolic mechanism of *Nigella sativa*:Anti-oxidant activity is thought to be responsible for the reduction in blood pressure in hypertensive patients: Thymoquinone, due to its structure, exhibits potent anti-radical scavenging activity, reducing ROS levels. *Nigella sativa* seed extracts have been shown to increase superoxide dismutase, catalase, and glutathione reductase activities. Collectively, these mechanisms result in a significant decrease in malondialdehyde (MDA), an indicator of lipid peroxidation [1,2,4,8,23]. The reduction in ROS levels is linked to the increased bioavailability of NO levels, which can lead to a reduction in blood pressure.Anti-inflammatory activity: Inflammation may play a role in cardiometabolic disease, so reducing inflammation can have a positive impact. *Nigella sativa* inhibits the inducible nitric oxide synthase and reduces the levels of TNF-α, IL-6, IL-1β, and other pro-inflammatory cytokines through the inhibition of the NF-κB signaling pathway. Additionally, *Nigella sativa* extracts have demonstrated the inhibition of cyclooxygenase 2 [8].Anti-dyslipidemic activities: *Nigella sativa* seed extracts inhibit the expression of HMG-CoA reductase while increasing the expression of LDL receptors [8]. These two activities result in a decrease in cholesterol levels.Anti-diabetic activity: Thymoquinone has been demonstrated to act as a PPAR-γ agonist, capable of improving insulin resistance. It can also inhibit intestinal α-glucosidase, reducing glucose absorption, and activate AMPK, increasing GLUT4 receptor availability while inhibiting hepatic gluconeogenesis. Thymoquinone, thanks to its anti-oxidant properties, can improve the integrity and proliferation of β pancreatic cells, enhancing insulin secretion.

### 4.1. Cardio-Protective and Anti-Hypertensive Activities

The cardio-protective effects of *Nigella sativa* seeds have been demonstrated in isoproterenol-induced myocardial injury in rats. It has been observed that the supplement exerts a protective effect on cardiac injury by mitigating oxidative stress and inflammation, improving anti-oxidant status, and preserving structural integrity [24] (Table 1).

It has been also reported that *Nigella sativa* seed administration (200 g of powder) improved cardiac functions in isolated rat hearts with ischemia–reperfusion injuries by attenuating oxidative stress [25] (Table 1).

Hypertension is one of the main modifiable risk factors for CVD. Some studies have shown the use of *Nigella sativa* seeds and has positive effects on the treatment of high blood pressure (Table 1).

It has been hypothesized that the anti-hypertensive effects of *Nigella sativa* may be related to its anti-oxidant, calcium channel blockade, and diuretic properties [23].

Hussain et al. demonstrated that the consumption of virgin *Nigella sativa* oil at the dose of 1 mL/day in 146 mild–moderate hypertensive and hyperlipidemic patients for 45 days significantly reduced systolic (SBP) and diastolic blood pressure (DBP) (*p* < 0.05) as compared to the baseline. Regarding the lipid profile, the supplement produced a significant decrease in total cholesterol (TC), low-density lipoprotein cholesterol (LDL-C), and triglycerides (Tg) and an increase in high-density lipoprotein cholesterol (HDL-C) (*p* < 0.05 for all) compared to initial values [26].

One study including 108 mild hypertensive patients who received 100 and 200 mg of *Nigella sativa* extract twice a day or placebo for 8 weeks, in which the two doses of supplement significantly reduced SBP and DBP (*p* < 0.01 for all) compared to placebo. In addition, the supplement decreased blood pressure values in a dose-dependent manner [27].

In another trial, 70 healthy volunteers with SBP from 110 to 140 mmHg and DBP from 60 to 90 mmHg were randomized to receive 5 mL/day *Nigella sativa* oil or placebo for 8 weeks. The supplement significantly decreased SBP (*p* = 0.002) and DBP (*p* = 0.040) as compared to placebo [28].

In the study by Shoaei-Hagh et al. [29], 55 hypertensive not well-controlled patients treated with thiazide, with or without angiotensin-converting enzyme (ACE) inhibitor or angiotensin receptor antagonist, were given 5 mL/day of *Nigella sativa* seed oil or placebo for 8 weeks. The supplement produced a significant decrease in SBP (*p* = 0.027) and DBP (*p* < 0.001) as compared to placebo. *Nigella sativa* seed oil also affected glycemic and lipid parameters significantly, reducing fasting plasma glucose (FPG) (*p* = 0.001), TC (*p* < 0.001), and LDL-C (*p* < 0.001) against the placebo. Moreover, a significant decrease was found in malondialdehyde (MDA) (*p* < 0.05) and an increase in glutathione reductase (*p* < 0.001) as compared to placebo. Daily supplementation of *Nigella sativa* seed oil in addition to anti-hypertensive agents seems to be an effective and safe mode for the complementary treatment of hypertension, with an accompanying improvement in lipid and glucose parameters.

A crossover, double-blind, placebo-controlled trial enrolled 39 obese and overweight women to evaluate the effect of *Nigella sativa* oil on cardiovascular risk factors including blood pressure, lipid profile, and atherogenic indices. These patients received the supplement at 2 g/day in the form of capsules or a placebo for 8 weeks. *Nigella sativa* oil significantly decreased SBP (*p* < 0.001) without affecting DBP. The supplement also reduced LDL-C (*p* = 0.031) and the TC/HDL-C ratio (*p* < 0.001), as an atherogenic index, whereas it increased HDL-C (=0.009) [30].

### 4.2. Anti-Diabetic Activity

Different studies reported that the possible hypoglycemic mechanisms of *Nigella sativa* were mediated by the activation of insulin and AMP-activated protein kinase (AMPK) pathways both in the muscle, where it promoted the synthesis and translocation of the glucose transporter 4 (GLUT4), and in the liver, where it inhibited gluconeogenesis. Furthermore, the anti-oxidant activity of *Nigella sativa* promoted the proliferation of β-pancreatic cells and their integrity, thus improving insulin secretion [31,32,33]. In addition, an in vitro study showed the inhibitory effect of *Nigella sativa* extract against intestinal α-glucosidase and pancreatic α-amylase, resulting in reduced bowel glucose absorption [34].

Some further studies demonstrated the potential anti-diabetic activity of *Nigella sativa* (Table 2).

One trial evaluated the effect of this supplement in 41 type 2 diabetes mellitus (T2DM) patients who drank *Nigella sativa* tea, prepared with one pack containing supplement seeds of 2.5 g, twice daily for 6 months in addition to their hypoglycemic drug therapy. A significant decrease in FPG, post-prandial glucose (PPG), and glycated hemoglobin (HbA1c) (*p* < 0.001 for all) was found compared to baseline values after supplementation [35].

A 3-month randomized study was conducted in 60 T2DM patients divided into three groups who received metformin 1 g/day and TQ 50 mg/day, metformin 1 g/day and TQ 100 mg/day, or metformin 1 g/day alone, respectively, for 3 months. The two doses of TQ in addition to a hypoglycemic drug were found to reduce HbA1c, FPG, and PPG to a greater extent than metformin monotherapy [36].

Another trial determined the effects of *Nigella sativa* oil extract on cardiometabolic risk factors in 43 T2DM patients who took the supplement in the form of soft gel capsules of 1 g/day or a placebo. After 8 weeks of treatment, *Nigella sativa* oil exerted beneficial effects, significantly reducing some parameters of glycemic control (FPG (*p* = 0.03) and HbA1c (*p* = 0.001)), the lipid profile (TC (*p* = 0.04), LDL-C (*p* = 0.001), and Tg (*p* = 0.003)), some anthropometric parameters (body mass index (BMI) and waist circumference (*p* < 0.001 for both)), and blood pressure values (SBP (*p* = 0.001) and DBP (*p* = 0.002)) [37]. Kooshki et al. evaluated the effects of *Nigella sativa* oil on inflammation parameters, oxidative stress, FPG, and the lipid profile in 50 T2DM patients. These patients were given the supplement as a capsule of 1 g/day or a placebo for 8 weeks. *Nigella sativa* produced a significant decrease in FPG (*p* < 0.001), the lipid profile (TC (*p* < 0.001), LDL-C (*p* < 0.001), and Tg (*p* < 0.001)), inflammation (highly sensitive C-reactive protein (hs-CRP) (*p* < 0.001)), and oxidative stress markers (MDA (*p* < 0.001)), as well as an increase in HDL-C (*p* < 0.001) as compared to the placebo, thus showing the cardiovascular protective effects in T2DM patients [38].

### 4.3. Anti-Obesity and Anti-Dyslipidemic Activities

Obesity or dyslipidemia has a linear relationship with cardiovascular and cerebrovascular diseases, resulting in an increased risk of mortality [2], and is also a predisposing factor for diabetes mellitus and cancer [39]. *Nigella sativa* has been studied to evaluate the use of medicinal plants in the treatment of obesity and dyslipidemia [2].

Some studies analyzed the potential mechanisms of the weight reduction effect of *Nigella sativa*, which include induction of anorexia or reduction in food intake and sensation of appetite [40,41], inhibition of intestinal glucose absorption [42], decrease in insulin secretion [40,43], and increase in adiponectin levels [44].

The weight reduction effect of *Nigella sativa* was investigated in several studies (Table 2).

A recent review included 14 randomized controlled trials (RCTs) whose study populations all had metabolic disorders such as diabetic patients, prediabetes, autoimmune hypothyroidism, and non-alcoholic fatty liver disease (NAFLD), or comprised obese and overweight individuals. All participants were treated with *Nigella sativa* oil or powder in the form of capsules. The review concluded that the administration of *Nigella sativa* at the dose of about 1–3 g/day for a period of 6–12 weeks was effective in lowering the body weight and other anthropometric parameters. It has been also shown that, at the same dosage, *Nigella sativa* oil might be superior to *Nigella sativa* powder in reducing body weight, probably due to the higher concentration of fatty acids and TQ [39].

In the study by Mostafa et al. [45], 105 obese prediabetic subjects were divided in three groups treated with a controlled diet and exercise regimen, with metformin at 1 g/day or *Nigella sativa* oil soft gelatin capsules at 900 mg/day, respectively, for 6 months. It was found that all study groups significantly reduced in their body weight (*p* < 0.001 for lifestyle modification and metformin groups; *p* = 0.002 for *Nigella sativa* group) and BMI (*p* < 0.001 for lifestyle modification and metformin groups; *p* = 0.002 for *Nigella sativa* group) as compared to baseline. As regards glycemic parameters, the lifestyle modification group showed only a significant increase in homeostatic model assessment of β-cell function (HOMA-B) (*p* < 0.001), whereas the metformin one exhibited a significant decrease in FPG (*p* < 0.001), PPG (*p* = 0.02) and homeostatic model assessment—insulin resistance (HOMA-IR) (*p* = 0.02) compared to baseline values. In the group treated with *Nigella sativa,* there was a significant reduction in FPG (*p* < 0.04), fasting plasma insulin (FPI) (*p* = 0.004), and HOMA-IR (*p* < 0.001) versus baseline. In terms of the lipid profile, only *Nigella sativa* significantly decreased TC, LDL-C, and Tg (*p* < 0.05), and increased HDL-C (*p* < 0.001) compared to the initial values. This supplement also affected inflammatory parameters, reducing tumor necrosis factor-α (TNF-α) (*p* = 0.001) as compared to baseline. In addition, *Nigella sativa* oil showed only mild gastrointestinal side effects and was better tolerated than metformin.

### 4.4. Anti-Oxidant and Anti-Inflammatory Activities

Oxidative stress and an increase in free radical level are the most important markers linked to several progressive pathological conditions, comprising neurological disorders, cancer, and endocrine diseases [46].

There are numerous medicinal plants with anti-oxidant effects, and among them, a more widely used one is *Nigella sativa*. This herb, as a potential source of natural anti-oxidants, reduces oxidative stress markers such as reactive oxygen species (ROS) and MDA, while upregulating anti-oxidant enzymes, such as superoxide dismutase (SOD) and catalase (CAT), and molecules, such as glutathione (GSH). The anti-oxidant activity can be attributed not only to TQ but also to other components of *Nigella sativa*, including phenolic compounds and flavonoids [47]. Several authors also showed that TQ can reduce nitric oxide (NO) production through inhibition of inducible nitric oxide synthase (iNOS) and then scavenge free radicals [48,49].

Concerning the potential mechanisms of *Nigella sativa* in systemic inflammation, these may be mainly due to TQ together with its carbonyl polymer (nigellone), thymol, thymohydoquinone, alpha-hederin, limonene, and polyphenols that reduce the production of inflammatory mediators such as 5- lipoxygenase, leukotriene, and eosinophils. It has been hypothesized that TQ can inhibit nuclear factor k chain transcription in the B cell (NF-κB) signaling pathway, and its transcription thus suppresses the expression of chemokines and pro-inflammatory cytokines such as TNF-α, IL-6, and IL-8 [50,51,52].

A recent systematic review and meta-analysis of 12 RCTs evaluated the effects of *Nigella sativa* on inflammatory and oxidative stress markers, including CRP, interleukin-6, TNF-α, total anti-oxidant capacity (TAC), and MDA. The targeted populations comprised patients with T2DM, obesity, NAFLD, ulcerative colitis, helicobacter infection, rheumatoid arthritis, and Hashimoto’s thyroiditis. Meta-analysis results showed that *Nigella sativa* consumption produced a significant reduction in CRP and MDA levels (*p* < 0.001 for both) and an increase in TAC (*p* = 0.01) without affecting TNF-α concentrations, and that the supplement might improve inflammation and oxidative status. However, this meta-analysis had some limitations, as reported by the authors. The included studies were heterogeneous in regard to study populations, types of *Nigella sativa* supplements, dosages, treatment durations, and methods for the measurement of inflammatory and oxidative markers. Furthermore, in the studies considered, mostly conducted in Iran, the amount of TQ present in the various preparations was not specified. All these factors prevent the anti-oxidant and anti-inflammatory activities of *Nigella sativa* from being clearly defined [53].

In the trial by Amizadeh et al. [54], the anti-inflammatory and anti-oxidant effects of *Nigella sativa* oil were analyzed in 89 Behcet’s disease patients who received 1 g/day of supplement or placebo for 8 weeks. It was observed that *Nigella sativa* produced a significant decrease in MDA (*p* = 0.015) and an increase in TAC (*p* = 0.003) without modifying interleukin-10, TNF-α, and hs-CRP levels with respect to the baseline (Table 2).

The anti-inflammatory and anti-oxidant activities of *Nigella sativa* oil were also studied to evaluate its therapeutic effectiveness on the parameters of disease activity, functional capacity, and radiological changes in 40 female patients with rheumatoid arthritis treated with methotrexate, hydroxychloroquine, folic acid, and diclophenac sodium. These patients took two capsules/day of placebo for 1 month and subsequently the supplement at 1 g/day for 1 month. *Nigella sativa* significantly decreased the disease activity score (DAS-28) (*p* = 0.017), the number of swollen joints (*p* < 0.0001), and the duration of morning stiffness (*p* = 0.016) compared to before and after the placebo [55] (Table 2).

**Table 2 biomedicines-12-00405-t002:** Summary on the anti-diabetic, anti-obesity, anti-dyslipidemic, anti-oxidant, and anti-inflammatory activities of *Nigella sativa*.

Experimental Model	Treatment and Period	Results	Reference
** *Anti-diabetic activity* **			
T2DM patients (*n* = 41)	5 g/day *Nigella sativa* seeds 6 months	FPG, PPG, and HbA_1c_ decrease	(El-Shamy 2011) [35]
T2DM patients (*n* = 60)	(1) Metformin 1 g/day + TQ 50 mg/day (2) Metformin 1 g/day + TQ 100 mg/day (3) Metformin 1 g/day 3 months	FPG, PPG, and HbA_1c_ decrease more than metformin monotherapy	(Ali 2021) [36]
T2DM patients (*n* = 43)	1 g/day *Nigella sativa* oil 8 weeks	BMI and waist circumference decrease SBP and DBP decrease FPG and HbA_1c_ decrease TC, LDL-C, and Tg decrease	(Hadi 2021) [37]
T2DM patients (*n* = 50)	1 g/day *Nigella sativa* oil 8 weeks	FPG decreases TC, LDL-C, and Tg decrease HDL-C increases hs-CRP decreases MDA decreases	(Kooshki 2020) [38]
** *Anti-obesity and anti-dyslipidemic activities* **			
Obese and prediabetic subjects (*n* = 105)	(1) Diet and exercise (2) Metformin 1 g/day (3) 900 mg/day *Nigella sativa* oil 6 months	BMI and body weight decrease Homa B increases (lifestyle modification group) FPG, PPG, and HOMA-IR decrease (metformin group) FPG, FPI, and HOMA-IR decrease (*Nigella sativa* group) TC, LDL-C, and Tg decrease and HDL-C increases (*Nigella sativa* group) TNF-decreases (*Nigella sativa* group)	(Mostafa 2021) [45]
** *Anti-oxidant and anti-inflammatory activities* **			
Behcet’s disease patients (*n* = 89)	1 g/day *Nigella sativa* oil 8 weeks	MDA decreases TAC increases	(Amizadeh 2020) [54]
Female patients with rheumatoid arthritis (*n* = 40)	1 g/day *Nigella sativa* oil 1 months	DAS-28, number of swollen joints, and duration of morning stiffness decrease	(Gheita 2012) [55]

T2DM: type 2 diabetes mellitus; NAFLD: non-alcoholic fatty liver disease; TQ: thymoquinone; FPG: fasting plasma glucose; PPG: post-prandial plasma glucose; HbA1c: glycated hemoglobin; BMI: body mass index; SBP: systolic blood pressure; DBP: diastolic blood pressure; TC: total cholesterol; LDL-C: low-density lipoprotein cholesterol; Tg: triglycerides; HDL-C: high-density lipoprotein cholesterol; hs-CRP: highly sensitive C-reactive protein; MDA: malondialdehyde; HOMA B: homeostatic model assessment of β-cell function; HOMA-IR; homeostatic model assessment—insulin resistance; FPI: fasting plasma insulin; TNF-α: tumor necrosis factor-α; TAC: total anti-oxidant activity; DAS: disease activity score.

### 4.5. Hepatoprotective Activity

Recent reviews underline the hepatoprotective effects of *Nigella sativa* and its bioactive components, such as TQ, thymol, and α-hederin [56,57].

The mechanisms underlying the hepatoprotective action of *Nigella sativa* and its constituents include (a) inhibition of lipid peroxidation and oxidative stress, (b) increase in GSH level and anti-oxidant enzymes, (c) reduction in fat accumulation, and (d) prevention of inflammation [2].

Liver function impairment may be caused by several factors, including hepatitis, steatosis, toxic chemicals, drugs, and radiation [58].

Fatty liver, also known as non-alcoholic fatty liver disease (NAFLD), is characterized by excessive accumulation of lipids, in particular, triglycerides, in the liver and includes a broad range of hepatic diseases, from fibrosis to non-alcoholic steatohepatitis (NASH). This condition over time can lead to liver cirrhosis or hepatocellular carcinoma [59,60,61]. NAFLD is also considered the hepatic manifestation of metabolic syndrome, which is associated with insulin resistance or T2DM, obesity, dyslipidemia, and hypertension [62].

Several studies have described the potential hepatoprotective effects of *Nigella sativa* extract or other preparations in this liver disease (Table 3).

A randomized controlled trial was performed on 70 patients with NAFLD who received *Nigella sativa* powder at 2 g/day or placebo for 3 months. A significant decrease in body weight (*p* = 0.041), BMI (*p* = 0.012) and both liver enzymes’ levels (*p* = 0.021 for AST and *p* = 0.036 for ALT) was observed after supplementation. Moreover, *Nigella sativa* improved the ultrasound grading of hepatic steatosis since 57.14% of patients had a normal fatty liver grading (*p* = 0.002), which was more when compared to the placebo [63].

Another study evaluated the effects of *Nigella sativa* and *Melissa officinalis* on hepatic enzymes and grade of fatty liver in 50 patients with NAFLD who were given a herbal tea consisting of intact seeds of *Nigella sativa* at 5 g and dry leaf powder of *Melissa officinalis* at 5 g or orlistat at 120 mg/day for 3 months. A significant decrease (*p* < 0.001) was found in AST (*p* < 0.001) and ALT (*p* < 0.006) as well as BMI as compared to baseline after supplementation. In addition, the association of *Nigella sativa* and *Melissa officinalis* significantly ameliorated the grade of fatty liver both compared to the baseline and placebo (*p* < 0.001) [64].

In the trials by Darand et al. [65,66], the effects of *Nigella sativa* seeds on inflammatory markers and CVD risk factors were analyzed in 41 patients with NAFLD who received a lifestyle modification and 2 g/day of supplement or placebo for 12 weeks. The supplement caused a reduction in hs-CRP (*p* = 0.0001) and NF-κB (*p* = 0.02) as compared to initial values, while TNF-α was significantly decreased both versus the baseline (*p* < 0.05) and placebo (*p* = 0.001). *Nigella sativa* seeds also significantly reduced serum glucose (*p* = 0.041), serum insulin (*p* = 0.027), and HOMA-IR (*p* = 0.021) and increased the quantitative insulin sensitivity check index (QUICKI) (*p* = 0.002) as compared to the placebo. However, the supplement did not affect the lipid profile. Moreover, a significant reduction was found in hepatic steatosis and its percentage (*p* < 0.005) compared to the baseline, whereas the steatosis percentage was reduced only versus the placebo (*p* = 0.005).

One study investigated the effects of *Nigella sativa* oil on the improvement in lipid and glycemic parameters, liver enzymes, and inflammatory markers in 44 patients with NAFLD who took 1 g/day of supplement or placebo. After 8 weeks of treatment, *Nigella sativa* significantly reduced FPG (*p* < 0.01), TC, LDL-C, Tg, very-low-density lipoprotein (VLDL) (*p* < 0.01 for all), AST, and ALT (*p* < 0.01 for both), whereas it increased HDL-C (*p* < 0.01) as compared to both to the baseline and placebo. As regards inflammatory markers, a decrease was observed in hs-CRP, TNF-α, and interleukine-6 (*p* < 0.01 for all) in comparison with both the baseline and placebo (*p* = 0.02 for hs-CRP, *p* < 0.01 for TNF-α, and *p* < 0.01 for interleukin-6) following supplementation. *Nigella sativa* did not change insulin, gamma-glutamyltransferase (γ-GT), or blood pressure versus either the initial values or the placebo [67].

Another trial evaluated the hepatoprotective effects of fully standardized *Nigella sativa* seed oil in 120 patients with NAFLD. These patients were given 5 mL/day of supplement or placebo for 3 months. A significant decrease was found in hepatic steatosis grade (*p* = 0.004), LDL-C (*p* = 0.01), Tg (*p* = 0.001), AST, and ALT (*p* < 0.001 for both) as well as an increase in HDL-C (=0.001) in the *Nigella sativa* group as compared to the placebo. No change was observed in blood urea nitrogen, creatinine, blood cell count, or BMI [68].

The anti-oxidant activity of *Nigella sativa* seed extract and its main bioactive molecule, thymoquinone, have a beneficial effect on liver function. This action leads to an increase in GSH levels and anti-oxidant enzymes, inhibition of fat accumulation, prevention of inflammation, and a reduction in cell damage. The reduction in TNF-α may be responsible for decreasing insulin resistance and, consequently, improving hepatic steatosis. Indeed, studies on NAFLD patients have demonstrated that supplementation with *Nigella sativa* seed extract improves inflammatory parameters such as TNF-α, hs-CRP, and IL-6 [65,67].

### 4.6. Effects of Nigella sativa in Menopause

Menopause is characterized by the end of menstrual cycles in the absence of any pathological or physiological cause for at least 12 consecutive months. Perimenopause, or menopausal transition, is a defined period of time beginning with the onset of irregular menstrual cycles until the final menstrual period (FMP) [69]. During this phase, women experience several early and late complications due to the gradual decline of the ovarian functions resulting in reduced levels of both estrogen and progesterone and an increase in follicle-stimulating hormone (FSH) concentration due to the decrease in inhibin-B secretion [70].

Some studies have investigated the effects of *Nigella sativa* in menopause (Table 4).

One trial evaluated the hypolipidemic effects of *Nigella sativa* in 37 menopausal women who received capsulate *Nigella sativa* seed powder at 1 g/day or placebo for 2 months. An improvement in the lipid profile was observed, with a decrease in TC, LDL-C. and Tg and an increase in HDL-C (*p* < 0.05 for all) as compared to the placebo after supplementation [71].

Another study assessed the effects of this supplement on metabolic syndrome in 30 menopausal women who were given *Nigella sativa* seed powder at 1 g/day or placebo for 2 months. The administration of the supplement significantly reduced FPG, TC, LDL-C, and Tg versus the placebo (*p* < 0.05 for all) [72].

The effects of *Nigella sativa* extract on metabolic syndrome were also investigated in 140 postmenopausal women who took the supplement in the form of a capsule at 500 mg/day or a placebo for 2 months. Similar to the above study, the supplement significantly reduced FPG, TC, LDL-C, and Tg as compared to the placebo (*p* < 0.001 for all) [73].

It has been observed that estrogens have anti-oxidant properties [74], which are related to their phenolic groups [75] and may also be beneficial in scavenging free radicals [76]. The reduced anti-oxidant activity of estrogen during menopause coincides with increases in ROS and some pathological conditions such as osteoporosis and menopausal symptoms [77].

As regards osteoporosis, Farshbaf-Khalili et al. [78] evaluated the effect of nanomicelle curcumin, *Nigella sativa* oil, and their association on the expression levels of specific microRNAs (miRNAs) in 120 postmenopausal women with low bone mass density (BMD). MicroRNAs are the key post-transcriptional repressors of gene expression and play an important role in regulating bone remodeling [79]. The expression levels of miR-21, miR-422a, and miR-503 were examined since they are mainly related to bone metabolism. In this study, curcumin was used since it exerts a protective effect on the bones via inhibition of osteoblast apoptosis [80] and osteoclast production [81], as well as induction of osteoclast death [82]. Postmenopausal women were randomly divided in four groups who received nanomicelle curcumin at 80 mg/day and placebo of *Nigella sativa*, *Nigella sativa* oil at 1 g/day and placebo of nanomicelle curcumin, nanomicelle curcumin at 80 mg/day and *Nigella sativa* oil at 1 g/day, or both placebos for 6 months, respectively. *Nigella sativa* oil alone and in combination with nanomicelle curcumin increased the expression levels of miR-21 (*p* = 0.037 for *Nigella sativa* oil; *p* = 0.043 for *Nigella sativa* oil with nanomicelle curcumin), attributing to this miRNA the potential role of inhibiting the progression of bone density loss using nutraceuticals such as *Nigella sativa* [78].

Moreover, in a previous study, low levels of miRNA-21 were detected in postmenopausal women with osteopenia and osteoporosis as compared to healthy women [83].

It has been shown that TQ is the main factor responsible for the action of *Nigella sativa* on osteoporosis [84].

An in vitro study reported that TQ plays a pivotal role in osteoblast differentiation and maturation. TQ’s effect on osteoblast maturation is associated with the increase in bone morphogenetic protein-2 (BMP-2) expression and activation of the extracellular signal-regulated kinase (ERK) pathway [85].

In addition, it has been observed that linoleic acid contained in *Nigella sativa* may positively benefit BMD in postmenopausal women [86].

As regards menopausal symptoms, several pharmaceutical and nonpharmaceutical interventions have been used [87]. However, many women often resort to complementary and alternative therapies to treat menopausal complications in a natural or safer way [88] instead of hormone replacement therapy (HRT) due to some of its side effects including increased risk of stroke, thromboembolic disorders, breast and endometrial cancers, adverse effects on lipids and lipase activity, and liver disorders [89].

There are several herbal products, such as *Cimicifuga racemose*, soy isoflavones, and *Trifolium pratense,* that are used as alternatives to pharmacological therapy [90].

Some studies showed the beneficial effects of *Nigella sativa* on menopausal symptoms. An open-label cross-over study examined the effect of 12-week use of encapsulated pure powdered *Nigella sativa* at 1.6 g/day in 41 perimenopausal women with climacteric symptoms. Treatment with this supplement significantly reduced the incidence and severity of menopausal symptoms (*p* < 0.05), as evaluated with the Greene Climacteric Scale (GCS) that independently measured psychological, anxiety, depression, somatic, and vasomotor symptoms. *Nigella sativa* also ameliorated some components of quality of life (*p* < 0.05) including general heath, role, vitality, and emotional and mental health, as evaluated with the SF-36 survey [91].

Another trial also reported that the daily administration of one *Nigella sativa* oil capsule (1 g) in 72 menopausal women for 8 weeks reduced the prevalence and severity of menopausal symptoms and the frequency of hot flashes (*p* < 0.001). However, the supplement had no significant effect on the serum levels of oxidative stress markers (TAC and MDA), despite the anti-inflammatory and anti-oxidant effects of *Nigella sativa* [92].

However, a study including 30 healthy postmenopausal women reported that a daily dose of 3 g of *Nigella sativa* powder significantly reduced MDA (*p* < 0.01) after 8 weeks of supplementation [93]. This incoherence is likely due to the different dosages of supplement used in the studies. In this regard, a study in rats fed with *Nigella sativa* oil at 10 mL/kg body weight for 8 weeks reported that the supplement significantly reduced MDA levels in liver tissue and the decrease was dose-dependent, since higher doses produced greater decreases in MDA levels [94].

In addition, the consumption of *Nigella sativa* powder at 1 g/day in 30 postmenopausal women for 2 months caused a significant decrease in body weight (*p* = 0.012) and BMI (*p* = 0.011) as compared to the baseline. The supplement also increased glutathione (GSH) levels (*p* < 0.001) versus initial values, indicating reduction in oxidative stress and contributing to a better quality of life in these women [95].

*Nigella sativa* powder supplementation at the dose of 1 g/day for 8 weeks in 30 healthy postmenopausal women significantly increased estradiol levels (*p* = 0.021) and reduced menopausal symptom severity, as evaluated with the Menopausal Rating Scale (MRS) subdivided into three categories, which assess somatic (hot flashes, heart discomfort, sleep problems, joint muscular discomfort) (*p* = 0.001), psychological (depressive mood, irritability, anxiety, physical and mental exhaustion) (*p* = 0.001), and urogenital (sexual problems, bladder problems, dryness of vagina) symptoms (*p* = 0.017) [96].

*Nigella sativa* supplementation could bring a broad-spectrum benefit to menopausal women who are not taking HRT due to preferences, side effects, or contraindications. Further studies are needed in larger samples to define both the potential mechanisms of action and the most effective phytochemical composition of this supplement.

In addition to the beneficial effects on lipids and insulin resistance through the mechanisms already described, an added benefit is protection of bone density. Thymoquinone affects osteoblast maturation through the increased expression of bone morphogenic protein-2 (BMP-2) and activation of the extracellular signal-regulated kinase (ERK) pathway. Furthermore, the improvement of psychological aspects related to climacteric syndrome could be linked to a reduction in cortisolemia, as recorded in a study of *Nigella sativa* seed oil. Cortisol is known to be linked to climacteric symptoms, as higher cortisol levels correspond to more severe symptoms.

**Table 4 biomedicines-12-00405-t004:** Summary of the effects of *Nigella sativa* in menopause.

Experimental Model	Treatment and Period	Results	Reference
Menopausal women (*n* = 37)	1 g/day *Nigella sativa* seeds powder 2 months	TC, LDL-C, and Tg decrease HDL-C increases	(Ibrahim 2014) [71]
Menopausal women with metabolic syndrome (*n* = 30)	1 g/day *Nigella sativa* seeds powder 2 months	FPG decreases TC, LDL-C, and Tg decrease	(Ibrahim 2014) [72]
Postmenopausal women with metabolic syndrome (*n* = 140)	500 mg/day *Nigella sativa* extract 2 months	FPG decreases TC, LDL-C, and Tg decrease	(Shirazi 2020) [73]
Postmenopausal women with low BMD (*n* = 120)	(1) 1 g/day Nigella sativa oil (2) 80 mg/day nanomicelle curcumin (3) 1 g/day Nigella sativa oil + 80 mg/day nanomicelle curcumin 6 months	MicroRNA-21 expression increases with *Nigella sativa* oil alone or in association with nanomicelle curcumin	(Farshbaf-khalili 2021) [78]
Perimenopausal women with climacteric symptoms (*n* = 41)	1.6 g/day *Nigella sativa* powder 12 weeks	Incidence and severity of menopausal symptoms decrease Quality of life (general health, role, vitality, emotional and mental health) improvement	(Latiff 2014) [91]
Menopausal women (*n* = 72)	1 g/day *Nigella sativa* oil 8 weeks	Prevalence and severity of menopausal symptoms decrease Frequency of hot flashes decreases	(Azami 2022) [92]
Healthy postmenopausal women (*n* = 30)	3 g/day *Nigella sativa* powder 8 weeks	MDA decreases	(Mostafa 2013) [93]
Postmenopausal women (*n* = 30)	1 g/day *Nigella sativa* powder 2 months	Body weight and BMI decrease GSH increases	(Sana 2019) [95]
Postmenopausal women (*n* = 30)	1 g/day *Nigella sativa* powder 8 weeks	Estradiol levels increase Menopausal symptom severity decreases	(Sana 2021) [96]

BMD: bone mass density; TC: total cholesterol; LDL-C: low-density lipoprotein cholesterol; Tg: triglycerides; HDL-C: high-density lipoprotein cholesterol; FPG: fasting plasma glucose; MDA: malondialdehyde; BMI: body mass index; GSH: glutathione.

## 5. Safety and Toxicity

Many studies have been conducted to evaluate the safety profile of *Nigella sativa* and TQ, recognized as the factor most responsible for its pharmacological activities.

Thymoquinone, as a quinone, is metabolized in vivo by cellular reductases to semiquinone (one reduction) or THQ (two reductions), which have pro-oxidant and anti-oxidant effects, respectively [97].

Various studies have shown that oral administration of TQ is safer intraperitoneally, having a higher LD50. The explanation may be that oral TQ is bio-transformed into less toxic metabolites in the gastrointestinal tract or metabolized in the liver into dihydrothymoquinone. Intraperitoneal administration of TQ, on the other hand, will lead to its complete absorption into the systemic circulation and, therefore, to an increase in toxicity [98].

Since TQ has lipophilic characteristics, in order to augment its pharmacological properties, Ong et al. [98] encapsulated TQ in a nanostructured lipid carrier (TQNLC) and evaluated TQ’s acute toxicity after oral administration of TQNLC and free TQ in mice. It was found that TQNCL is less toxic than pure TQ. The oral LD50 of TQ was 50–300 mg/kg, as estimated using the Globally Harmonized System (GHS) of Classification and Labelling of Chemicals, and thus TQ was classified in category 3 of the GHS. The LD50 of TQNLC was 300–2000 mg/kg and it was classified in category 4 of the GHS. The encapsulation of TQ in a nanostructured lipid carrier reduced TQ toxicity and seems to be a promising strategy for its clinical applications. In subacute toxicity evaluations, the administration of TQ or TQNLC had a No Observed Adverse Effect Level (NOAEL) of 10 mg/kg/day in mice.

In another study, a nanoemulsion rich in TQ, formulated to enhance its absorption and bioavailability, was shown to have no adverse side effects when administered at a dose of 44.5 mg TQ/kg in rats for 14 days [97,99].

*Nigella sativa* is approved for food use and Generally Recognized as Safe (GRAS) by both the US Food and Drug Administration (FDA) and the Flavor and Extract Manufacturers Association (FEMA). The European Food Safety Authority (EFSA) considers the essential oil a “chemical of concern”, due to the presence of certain alkaloids (Burdock GA Assessment of black cumin (*Nigella sativa* L.)), when regard to its use as a food ingredient and putative therapeutic agent [3].

Acute and sub-chronic toxicity studies in rats have assessed the safety of a *Nigella sativa* oil formulation containing 5% TQ (BCO-5) and established that the safe human dosage may be a maximum of TQ 48.6 mg/day [100].

Recently, the safety of BCO-5 at a dose of 200 mg/day for 3 months was evaluated in 70 healthy subjects. No significant changes were observed in the biochemical parameters related to liver (ALT, AST, alkaline phosphatase (ALP)) and renal (serum creatinine and urea) function. However, a significant decrease was found in TC (−22.32 mg/dL, −12.10%, *p* < 0.05), LDL (−18.59 mg/dL, −16.33%, *p* < 0.05), HDL (+6.45 mg/dL, +15.27%, *p* < 0.05), and Tg (−26.43 mg/dL, −19.66%, *p* < 0.05) as compared to the baseline, but within the normal range. No serious adverse events were reported during the trial. Bloating on the first days, mild diarrhea, borborygmi, and burping with a taste of *Nigella sativa* oil in the mouth throughout the intervention were registered [101].

Other studies showed the safety and tolerability of *Nigella sativa* oil using higher dosages. In one trial, 72 T2DM patients were given 3 g/day of *Nigella sativa* oil containing 14.5% TQ or placebo for 12 weeks. The supplement improved the glycemic and lipid profiles without side effects during the study except for mild gastrointestinal disorders [102].

Another study enrolled 50 obese women who received a low-calorie diet with 3 g/day of *Nigella sativa* oil containing 12.5% TQ or a low-calorie diet with the placebo for 8 weeks. The supplement had a positive effect on obesity, in reducing weight and increasing superoxidase dismutase levels, without any side effects being reported [103].

In a randomized, open-label, prospective, three-arm, parallel, multicenter study, 60 T2DM patients took metformin at 1 g/day with TQ at 50 mg/day, metformin at 1 g/day with TQ at 100 mg/day, or metformin at 1 g/day, respectively, for 3 months. It was observed that TQ as an adjunct therapy to a conventional drug resulted in better glycemic control compared to metformin alone. None of the patients treated experienced serious adverse events. The adverse events were diarrhea, epigastric pain, abdominal pain, and stomach ache. The study suggested that TQ add-on therapy with metformin was safe and well-tolerated in T2DM patients [36].

## 6. Pharmaceutical Technology Applied to *Nigella sativa*

The TQ content in *Nigella sativa* seeds and in its extracted oil varies according to the geographical region of origin, which determines the quantities of phytoconstituents. In this regard, the amount of TQ contained in Iranian-derived *Nigella sativa* essential oil is 13.7%, whereas its amount in the Indian-derived essential oil is up to 50% [104]. The TQ concentration in *Nigella sativa* oil from the Marche region cultivar in Italy is higher than those present in other *Nigella sativa* oils produced in the Middle East and in other Mediterranean regions [4].

Besides geographical variation, the content of TQ is influenced by the method of extraction used [104].

The objective of the extraction techniques adopted over time has been to obtain a pure volatile oil with a high concentration of TQ. Traditional methods like steam distillation and hydrodistillation allow us to isolate the volatile oil directly from the seeds but result in a significant loss of TQ [105]. A two-stage extraction method was developed and was found more efficient in conserving the TQ content in the volatile oil fraction of *Nigella sativa* [105,106]. The first stage consists of the isolation of the whole oleoresin from the seeds using mechanical expression or hexane extraction. The second one includes the hydrodistillation or steam distillation of the obtained oleoresin in order to isolate the pure volatile oil fraction. Other extraction methods that are performed include the use of a (a) solvent, (b) cold press, or (c) supercritical fluid extraction (SFE) [104]. As regards solvent extraction, this may produce detectable traces of organic solvent within the oil as well as induce the oxidation and degradation of the desired components. Cold press extraction has an oil content yield of 10–12%; that is significantly lower than that obtained with SFE, which can produce 4–5 times higher concentrations of pharmacologically active components, especially TQ [104]. However, when SFE was performed at high extraction pressure (150–300 bar), the obtained oleoresin comprised mainly the lipid fraction of seeds rather than the volatile oil fraction [106]. Other advanced methods include ultrasound-assisted extraction (UAE) and microwave-assisted extraction (MAE) [107].

A recent extraction method is the subcritical CO_2_ extraction technique (SbFE) that uses liquid CO_2_ as a green solvent to obtain, under mild operating conditions (70 bar, 30 °C), a volatile oil-rich fraction from the seeds of *Nigella sativa* with a minor lipid amount. Gas chromatography–mass spectrometry (GC-MS) analysis showed that the volatile oil obtained from SbFE contains the highest TQ content (60.5%) as compared to that obtained from hexane extraction at room temperature (37.6%) or at 60 °C (41.6%), hydrodistillation (10.2%), or steam distillation (23.7%). The volatile oil-rich oleoresin with a high TQ content from *Nigella sativa* seeds after SbFE can be applied in pharmaceutical and nutraceutical products [107].

The extraction technique utilized may influence not only the volatile components of *Nigella sativa* oil, in particular, the TQ content, but also the physiochemical properties and the fatty acid concentration [108].

## 7. Legislative Aspects

As previously reported, *Nigella sativa* is approved for food use and is Generally Recognized as Safe (GRAS) by both the US Food and Drug Administration (FDA) and the Flavor and Extract Manufacturers Association (FEMA). The European Food Safety Authority (EFSA) considers the essential oil a “chemical of concern” due to the presence of certain alkaloids (e.g., nigellimine) [3].

In 2021, the EFSA defined black cumin as a non-novel food, since it has been present on the market as a food or food ingredient and consumed to a significant degree before 15 May 1997, meaning it is exempt from the Novel Food Regulation (EU) 2015/2283 [109].

In Italy, *Nigella sativa* seeds and its oil are covered in annex 1 of the decree of 9 January 2019 that replaced annex 1 of the ministerial decree of 10 August 2018 relating to the “Discipline of the use of plant substances and preparations in food supplements”, as well as annex 1 of the ministerial decree of 9 July 2012 and annex 1 bis (BELFRIT list) introduced by the decree of 27 March 2014. Annex 1 of the decree of 9 January 2019 regulates the use of herbal substances and preparations in food supplements. They are listed in the document describing the plants and parts of them allowed in the composition of food supplements, together with the physiological effects recognized for each of these and any warnings. Concerning *Nigella sativa*, the seeds, and the oil extracted from them, the only warning is as follows: “not recommended for children and adolescents, pregnant and breastfeeding women”.

## 8. The International Market for Products Based on *Nigella sativa*

Commercial products based on *Nigella sativa* oil rarely report the TQ content, making their therapeutic use difficult. A study conducted in Malaysia, one of the countries where this natural remedy is historically known, highlights the heterogeneity of the TQ concentration of *Nigella sativa* oil products. In this study, 10 products on the Malaysian market were selected and their TQ content was measured using an established high-performance liquid chromatography (HPLC) method. The TQ content in the commercial *Nigella sativa* oil products ranged from 0.07% wt/wt to 1.88% wt/wt with a 27-fold difference between the lowest and the highest concentration. This difference was mainly due to the source of *Nigella sativa* oil as well as the extraction method, as previously described. Since the therapeutic effect of commercial *Nigella sativa* oil products is linked to the TQ content, some of them may not provide the TQ amount needed for some diseases; to remedy this, the regulation of its content in commercial *Nigella sativa* oil products was recommended by the authors [110].

Similar results have been reported by Khaikin et al. [111], who evaluated the quality of products based on *Nigella sativa* oil (bottled oil or oil capsules) available on the Swiss market in terms of TQ concentration, identifying one that was suitable for a clinical study. The methods of analysis used were methanolic extraction and HPLC-UV. All 11 products analyzed (6 bottled oils and 5 oil capsules) originated from Egypt, except for one from India; they were produced in the form of capsules in Switzerland and Australia and are available in these countries. It was observed that the TQ content varied greatly, at between 0.003% and 0.8% wt/wt. As in the previous study, the authors suggested that the TQ content should be declared on all *Nigella sativa* oil products in order to allow clinicians to choose an appropriate study medication and to allow consumers to buy products containing sufficient TQ amounts for the maintenance and improvement of their health.

A *Nigella sativa* oil formulation containing 5% TQ (BCO-5; patented and registered as BlaQmax^®^) is available on the Indian market and exhibits significant clinical efficacy to alleviate sleep disorders and stress [112].

It was also reported that BCO-5 was safe when supplemented in healthy subjects at a dose of 200 mg/day for 3 months [101].

## 9. The Italian Market for Products Based on *Nigella sativa*

In our country, there are several products based on black cumin oil. One of the preparations belonging to the herbal tradition is the mother tincture of *Nigella sativa*, obtained from the seeds. The pharmacological activities attributed to this preparation are anti-histamine, choleretic, carminative, emmenagogue, and diuretic, whereas the therapeutic indications are for a migraine, headache, flatulence, dyspepsia, bronchospasm, amenorrhea, and dysmenorrhea [113,114]. There are also other products in the form of black cumin oil and oil capsules, consumed for their content in fatty acids, such as linoleic acid, or for the minerals, such as iron. The TQ content is not reported on any of these products.

There are several products for external use, which exploit the moisturizing properties of black cumin oil and the ability to counteract hair loss.

The only product available for the Italian market titrated in TQ is Nisatol^®^, marketed by PharmExtracta (Pontenure, Italy). This food supplement comes in soft gel capsules containing *Nigella sativa* seed oil, titrated to 10% in TQ. The oil is obtained via the SbFE method, at temperatures and pressures suitable to obtain that final concentration of TQ, starting from a much less concentrated oil, obtained after cold pressing *Nigella sativa* seeds, with an initial titer of TQ < 0.5%. Each Nisatol^®^ soft gel contains 400 mg of *Nigella sativa* seed oil titrated to 10% TQ (40 mg), along with 3.3 mg of vitamin E in order to preserve the oil from oxidation. It has been shown that Nisatol^®^ can be useful in the treatment of menopausal women with metabolic syndrome due to the beneficial effects on both the glycemic and lipid profiles and on body weight, as well as for the psychological symptoms of menopause, bone mineral density, and blood pressure [103,115].

Furthermore, Nisatol^®^ can be used as an adjunctive therapy in the drug treatment of hyperglycemia, dyslipidemia, hypertension, NAFLD/NASH, and diabetic female sexual dysfunction [116], if the standard treatment is not enough to reach the desired outcomes.

Regarding innovative technologies or formulations that significantly improve the bioavailability of *Nigella sativa* compounds, a standardized *Nigella sativa* seed oil extract, highly titered in thymoquinone, is thought to be the supplement that should be used in future investigations. This specific extract, titered at 10% thymoquinone, allows for a good quantity of thymoquinone in a small amount of oil, reducing gastrointestinal side effects that can occur with higher oil amounts.

## 10. Conclusions

Evidence from the literature shows that the parts of *Nigella sativa* mainly used are its seeds, consumed whole, chopped, or subjected to pressing to obtain an oil, which can undergo subsequent technological processes. The remarkable and growing interest in this remedy is evidenced by several studies aimed at identifying rational clinical uses in different fields. Many pharmacological properties have been attributed to *Nigella sativa* and to the main bioactive constituent, TQ, which contribute to their potential health benefits against a wide range of disease conditions. Thymoquinone is a small molecule capable of interacting with several biological systems in different tissues. This multi-targeting activity explains the interest in *Nigella sativa* in diverse clinical contexts, since it can act on several aspects of the same condition, as in the case of menopausal women with metabolic syndrome.

However, the analysis of the literature revealed that the clinical use of *Nigella sativa* is limited by the lack of phytochemically characterized preparations, which curb the reproducibility and reliability of the results. Some studies have underlined both the great shortage, in Italy and abroad, of standardized preparations titrated in TQ, which implies uncertainty in using *Nigella sativa* oil, regarding results and safety, and the need to improve these in the future. In this regard, the scientific community should endeavor to optimize and standardize these preparations with high titers of TQ for the future development of *Nigella sativa*-based potential therapeutic agents, to be applied for various diseases. The scientific literature is unanimous in identifying thymoquinone as the main active ingredient for *Nigella sativa*’s activities, despite the fact that its seeds are rich in bioactive molecules. Different seed extracts of different types are used in the various studies cited, but considering the importance of thymoquinone, it seems necessary that its content should always be known when using a *Nigella sativa* preparation. Menopausal women have an increased risk of developing metabolic syndrome and, consequently, cardiovascular disease. Recent data have shed light on the strong connection between climacteric syndrome and the risk of cardiovascular disease [117,118]. In this area, *Nigella sativa* seed oil, given its broad spectrum of activities addressed in this work, can offer a useful tool to prevent metabolic syndrome, especially during the menopausal period. However, there are few studies that have investigated the efficacy of *Nigella sativa* extract in this population; further and more thorough studies are needed to understand the full potential of *Nigella sativa* seed oil.

## Figures and Tables

**Figure 1 biomedicines-12-00405-f001:**
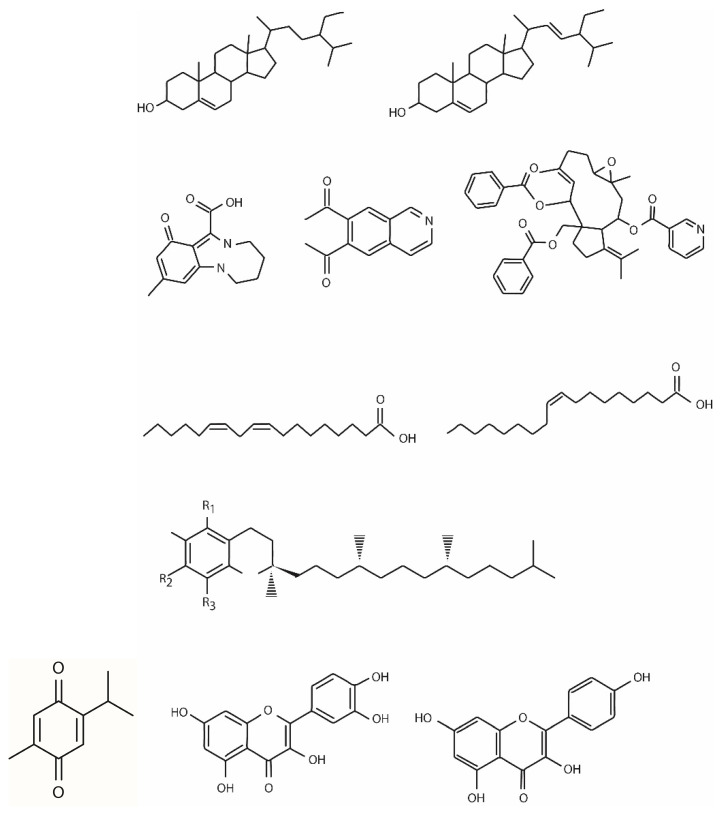
Chemical structure of thymoquinone and other chemical constituents of *Nigella sativa* L.

**Table 1 biomedicines-12-00405-t001:** Summary of the cardio-protective and anti-hypertensive activities of *Nigella sativa*.

Experimental Model	Treatment and Period	Results	Reference
** *Cardio-protective activity* **			
Isoproterenol-induced myocardial infarction in rats	TQ (20 mg/kg) 21 days	Mitigation of oxidative stress and inflammation Improvement of anti-oxidant status Preservation of structural integrity	(Ojha 2015) [24]
Rats with ischemia–reperfusion injury	200 g of powder	Improvement of oxidative stress	(Ghoreyshi 2020) [25]
** *Anti-hypertensive activity* **			
Mild–moderate hypertensive and hyperlipidemic patients (*n* = 146)	1 mL/day *Nigella sativa* virgin oil 45 days	SBP and DBP decrease TC, LDL-C, and Tg decrease HDL-C increases	(Hussain 2017) [26]
Mild hypertensive patients (*n* = 108)	100–200 mg twice day *Nigella sativa* extract 8 weeks	SBP and DBP decrease in a dose-dependent manner	(Dehkordi 2008) [27]
Healthy volunteers (SBP ≥ 110 and ≤140 mmHg; DBP ≥ 60 and ≤90 mmHg) (*n* = 70)	5 mL/day *Nigella sativa* oil 8 weeks	SBP and DBP decrease	(Fallah Huseini 2013) [28]
Hypertensive not well-controlled patients (*n* = 55)	5 mL/day *Nigella sativa* seeds oil 8 weeks	SBP and DBP decrease FPG, TC, and LDL-C decrease MDA decreases Glutathione reductase increase	(Shoaei-Hagh 2021) [29]
Obese and overweight women (*n* = 39)	2 g/day *Nigella sativa* oil 8 weeks	SBP decreases LDL-C and TC/HDL-C ratio decrease HDL-C increases	(Rampoosh 2021) [30]

SBP: systolic blood pressure; DBP: diastolic blood pressure; TQ: thymoquinone; TC: total cholesterol; LDL-C: low-density lipoprotein cholesterol; Tg: triglycerides; HDL-C: high-density lipoprotein cholesterol; FPG: fasting plasma glucose; MDA: malondialdehyde.

**Table 3 biomedicines-12-00405-t003:** Summary of the hepatoprotective activities of *Nigella sativa*.

Experimental Model	Treatment and Period	Results	Reference
NAFLD patients (*n* = 70)	2 g/day *Nigella sativa* powder 3 months	BMI and body weight decrease AST and ALT decrease Ultrasound grading of hepatic steatosis improves	(Hussain 2017) [63]
NAFLD patients (*n* = 50)	(1) 5 g/day *Nigella sativa* seeds + 5 g/day *Melissa officinalis* dry leaf powder (2) Orlistat 120 mg/day 3 months	BMI decreases AST and ALT decrease Grade of fatty liver improves	(Hosseini 2018) [64]
NAFLD patients (*n* = 41)	2 g/day *Nigella sativa* seeds 12 weeks	Serum glucose, serum insulin, and HOMA-IR decrease QUICKI increases hs-CRP, NF-κB, and TNF-α decrease Hepatic steatosis and its percentage decrease	(Darand 2019 a; Darand 2019 b) [65,66]
NAFLD patients (*n* = 44)	1 g/day *Nigella sativa* oil 8 weeks	FPG decreases TC, LDL-C, Tg, and VLDL decrease HDL-C increases AST and ALT decrease hs-CRP, TNF-α, and interleukine-6 decrease	(Rashidmayvan 2019) [67]
NAFLD patients (*n* = 120)	5 mL/day *Nigella sativa* seed oil fully standardized 3 months	LDL-C and Tg decrease HDL-C increases AST and ALT decrease Hepatic steatosis grade decreases	(Khoche 2019) [68]

NAFLD: non-alcoholic fatty liver disease; BMI: body mass index; AST: aspartate aminotransferase; ALT: alanine aminotransferase; HOMA-IR; homeostatic model assessment—insulin resistance; QUICKI: quantitative insulin sensitivity check index; hs-CRP: highly sensitive C-reactive protein; NF-κB: nuclear factor k chain transcription in B cells; TNF-α: tumor necrosis factor-α; FPG: fasting plasma glucose; TC: total cholesterol; LDL-C: low-density lipoprotein cholesterol; Tg: triglycerides; VLDL: very-low-density lipoprotein; HDL-C: high-density lipoprotein cholesterol.

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
