# Peer review of "The Use of Nigella sativa in Cardiometabolic Diseases"

_biomedicines, 2024, doi:10.3390/biomedicines12020405_

Round 1

Reviewer 1 Report

Comments and Suggestions for Authors

The review article titled “The use of nigella sativa in cardiometabolic diseases”.  However, the manuscript's emphasis does not seem to prioritize novelty, as it predominantly presents well-established information. Furthermore, the review lacks up-to-date information. Hence, I cannot recommend this manuscript for publication at this present information.

Some critical comments and suggestions

1.      The abstract and introduction are too general. The authors failed to focus on the title of the review and suggested updating those sections accordingly.

2.      Why is the only chemical structure presented in Figure 1? It should be presented with the remaining molecules.

3.      Moreover, those presented biomolecules do not focus on “cardiometabolic diseases”. If not have an activity no need to present the details.

4.      Section. 4. The protective cardiometabolic mechanism of nigella sativa should be illustrated.

5.      Further, the mechanism should be illustrated for the remaining major section or diagrammatic representations need to be presented.

6.      Contraction between the sections. For example, some of the sections discussed the phytochemicals that exist in nigella sativa but most of them particularly, the tables mentioned the plant extracts. The author should be clearer about what they going to deliver to the readers from this review.

Comments on the Quality of English Language

Minor editing of English language required

Author Response

Question:  The abstract and introduction are too general. The authors failed to focus on the title of the review and suggested updating those sections accordingly.

Answer: We did as you suggested.

Question: Why is the only chemical structure presented in Figure 1? It should be presented with the remaining molecules.

Answer: The scientific community agrees that Thymoquinone is the most prominent constituent of Nigella sativa seed extract and is widely acknowledged as the primary contributor to the pharmacological activities of Nigella sativa. In Fig. 1, the chemical structures of the most important molecules for each family are illustrated.

Question: Moreover, those presented biomolecules do not focus on “cardiometabolic diseases”. If not have an activity no need to present the details.

Answer: the other biomolecules are briefly described to provide a comprehensive characterization of Nigella sativa phytochemistry. These bioactive molecules might contribute to the cardiometabolic protection of Nigella sativa.

Question: The protective cardiometabolic mechanism of nigella sativa should be illustrated.

Answer: We did as you suggested.

Question: Further, the mechanism should be illustrated for the remaining major section or diagrammatic representations need to be presented.

Answer: We did as you suggested.

Question: Contraction between the sections. For example, some of the sections discussed the phytochemicals that exist in nigella sativa but most of them particularly, the tables mentioned the plant extracts. The author should be clearer about what they going to deliver to the readers from this review.

Answer: We did as you suggested.

Reviewer 2 Report

Comments and Suggestions for Authors

Present please the chemical structures of the compounds discussed in the article

Elaborate on the specific mechanisms through which thymoquinone exerts its antioxidant and anti-inflammatory effects in the human body

Based on your findings, what areas within Nigella sativa research do you believe require further exploration? Are there specific health conditions or diseases where Nigella sativa shows promising potential?

Did you come across any innovative technologies or formulations that significantly improved the bioavailability of Nigella sativa compounds?

Comments on the Quality of English Language

The quality of English is OK, however there are some syntax and grammar mistakes in the text

Author Response

Question: Present please the chemical structures of the compounds discussed in the article

Asnwer: We did as you suggested.

Question: Elaborate on the specific mechanisms through which thymoquinone exerts its antioxidant and anti-inflammatory effects in the human body

Answer: We did as you suggested.

Question: Based on your findings, what areas within Nigella sativa research do you believe require further exploration? Are there specific health conditions or diseases where Nigella sativa shows promising potential?

Answer: we did as you suggested.

Question: Did you come across any innovative technologies or formulations that significantly improved the bioavailability of Nigella sativa compounds?

Answer: We did as you suggested.

Round 2

Reviewer 1 Report

Comments and Suggestions for Authors

-